# Predicting Sensation-Seeking from Resting-State fMRI: The Need for Age-Specific Models

Zishen Li[1], Bishal Lamichhane[1], Ankit Patel[1,2], Ramiro Salas[2], Nidal Moukaddam[2], and Ashutosh Sabharwal[1]

*Abstract*—Sensation-seeking, as a sub-dimension of impulsivity, reflects an individual's tendency for novel and stimulating experiences. High sensation-seeking often involves novelty-seeking and risk-taking, which may lead to risky behaviors such as reckless driving, addiction, and substance use, significantly impacting individuals' social and personal functioning. Recent studies have utilized functional magnetic resonance imaging (fMRI) to study the neural mechanisms of sensation-seeking. However, the influence of demographic factors like age on the neural patterns associated with sensation-seeking remains unexplored. In this paper, we predicted sensation-seeking scores from resting-state fMRI data in a large-scale study involving 131 male participants aged 20 to 79. By developing separate predictive models for different age groups (age-specific model), we achieved an $R^2$ of 0.38 between the actual and predicted sensation-seeking scores. The proposed age-specific model significantly outperformed the baseline that fitted a single model for the entire dataset, indicating the importance of demographic factors in understanding the neural correlates of sensation-seeking. We identified key brain regions associated with sensation-seeking, including the prefrontal areas, cerebellum, subcortical regions, parietal lobe, and cerebral cortex areas. Notably, the brain connectivity patterns linked to sensation-seeking varied across age groups, further demonstrating the age-related variation in neural correlates of sensation-seeking. Our proposed age-specific modeling of sensation-seeking acknowledges the diversity in neural patterns across different aging stages and potentially offers more accurate insights into the neural correlates of sensation-seeking.

*Index Terms*—rs-fMRI, Function Connectivity, Sensation-seeking, Age-specific modeling

## I. INTRODUCTION

Impulsivity is a multifaceted construct characterized by the tendency to act on impulse without considering the consequences, often leading to unplanned actions. Sensation-seeking, a sub-dimension of impulsivity, is characterized by the tendency to seek for novel and thrilling experiences [1]. While sensation-seeking can sometimes drive creativity and exploration, it is often linked to risky behaviors such as reckless driving, alcohol dependence, risky sexual behaviors, and mental health conditions, including substance use disorders and psychopathic personalities [1], [2]. Understanding sensation-seeking is crucial for developing interventions and treatments for these conditions. Assessment of sensation-seeking generally includes self-reported surveys and behavioral assessments. Self-reported questionnaires include the Zuckerman's Sensation Seeking Scale (SSS) [3], the Impulsive Sensation Seeking Scale (ImpSS) [4], and the UPPS-P Impulsive Behavior Scale [5], which measure sensation-seeking either on its own or as a sub-dimension of impulsivity. Behavioral measures include various laboratory tasks such as the Go/No-Go task, Stop-Signal Task, Balloon Analogue Risk Task (BART), and Iowa Gambling Task (IGT), which assess risk-taking behavior and response inhibition that correlates with sensation-seeking tendencies [6], [7].

Many recent studies have investigated neuroimaging-based approaches as an objective assessment of sensation-seeking. Unlike subjective measurements such as questionnaires and behavioral tests, neuroimaging approaches allow for direct investigation of the neural mechanisms of sensation-seeking that may help develop potential interventions or treatments. Functional magnetic resonance imaging (fMRI) is a commonly used neuroimaging technique to study brain activities. Previous studies have employed fMRI to identify specific brain structures involved in sensation seeking [8]. For example, previous studies have shown increased fMRI activations in the bilateral ventral striatum, bilateral thalamus, and cerebellum during reward expectation in healthy subjects—a characteristic often associated with sensation-seeking [9]. High sensation seekers exhibit notable differences in brain activity compared to low sensation seekers, particularly in the prefrontal cortex and anterior insula areas, in response to reward [10]. Additionally, high sensation seekers show greater insula and posterior medial orbitofrontal cortex responses to arousing stimuli, which reflects their tendency for novelty-seeking [11]. Event-related potential (ERP) studies revealed that high sensation seekers show greater responses to novel objects in the N2 ERP component observed over frontal brain regions, correlating with fMRI responses in the orbitofrontal gyrus [12]. Sensation-seeking is also linked to deficits in response inhibition, with decreased activation in the right inferior frontal gyrus, prefrontal cortex, anterior cingulate cortex, and anterior lateral orbitofrontal cortex [13]–[15]. In addition to related brain activations and connectivity, previous studies have attempted to predict sensation-seeking from brain imaging data. For instance, Wan *et al.* predicted sensation-seeking from brain functional connectivity with a correlation coefficient $r = 0.34$ between the predicted and actual values. The authors also identified significant connections between the medial orbitofrontal cortex and the anterior cingulate cortex in predicting sensation-seeking [16]. However, the study proposed a single prediction model for the entire population, overlooking the variations in brain development and connectivity across different aging phases [17], which could be crucial for improving prediction accuracy and understanding the neural correlates of sensation-

[1] Department of Electrical and Computer Engineering, Rice University, Houston, TX, USA

[2] Baylor College of Medicine, Houston, TX, USA

seeking.

To address the gap in previous studies regarding the potential age-related differences in how brain activity relates to sensation-seeking, we investigated age-specific prediction models to improve the predictive accuracy of sensation-seeking. We evaluated the modeling on the 131 male participants from the MPI Leipzig Mind-Brain-Body (LEMON) dataset. The male subset was selected for its larger sample size and greater variability in sensation-seeking scores compared to the female population in the dataset. This focus allowed us to investigate age-related differences without the confounding effects of gender-associated differences (investigating the multi-demographic factors requires an even larger and more gender-balanced sample, and should be considered in future work). Through unsupervised methods, we observed distinct age-related clusters of functional connectivity features, indicating varied functional connectivity distributions across age groups. This variation suggested that the associations between functional connectivity and sensation-seeking were not uniform across ages. Therefore, we proposed an age-specific modeling approach to capture these distinct brain connectivity patterns. As a baseline, we used a single prediction model across all ages, following prior research methodologies [16]. Our results demonstrated that the age-specific model outperformed the single all-ages model. By capturing the distinct brain connectivity patterns present in different age groups, we achieved better prediction and understanding of sensation-seeking traits.

## II. MATERIALS AND METHODS

### A. Data

*1) Participants:* The dataset used in this study is part of the large MPI Leipzig Mind-Brain-Body (LEMON) dataset designed to provide a comprehensive resource for studying the complex relationships between mind, brain, and body functions [18]. The dataset comprises multi-modal data, including brain imaging data (MRI and EEG), cognitive assessments, emotional measures, and peripheral physiology data collected from 227 healthy participants. Participants were recruited at the University of Leipzig, Germany, and the data collection protocol was in accordance with the Declaration of Helsinki. The participants fell into two age groups: a younger group aged $20 - 35$ ($N = 153$, 108 males) and an older group aged $59 - 77$ ($N = 74$, 37 males). We excluded the participants with incomplete sensation-seeking scores, and the final dataset comprises 205 individuals, with 139 from the young age group (aged $20 - 35$, 96 males) and 66 from the old age group (aged $59 - 77$, 35 males).

*2) Self-reported sensation-seeking score:* The LEMON dataset measures sensation-seeking using the UPPS impulsivity questionnaire [19]. UPPS evaluates the level of impulsivity across four sub-dimensions:

- Urgency: This sub-dimension describes the impulsive tendency triggered by intense emotions.
- Lack of Premeditation: This refers to acting without considering consequences beforehand.

- Lack of Perseverance: This involves difficulty in maintaining focus on tasks.
- Sensation-Seeking: This reflects a preference for stimulation and excitement, often leading to risky behaviors.

Participants rated 45 questionnaire items on a 4-point Likert scale (1 = strongly agree to 4 = strongly disagree). Each item relates to one of the impulsivity sub-dimensions, and the aggregated score is used to assess impulsivity levels across these sub-dimensions.

Figure 1 illustrates the distribution of sensation-seeking scores across different age and gender groups. Notably, sensation-seeking scores decrease with age, and males tend to exhibit higher levels of sensation-seeking than females, which aligns with previous literature [20]. Given the larger sample size of males within each age range and the wider dynamic range of sensation-seeking scores, our study focused on examining age-related differences in the neural correlates of sensation-seeking among the male population, remaining insensitive to possible gender-related differences.

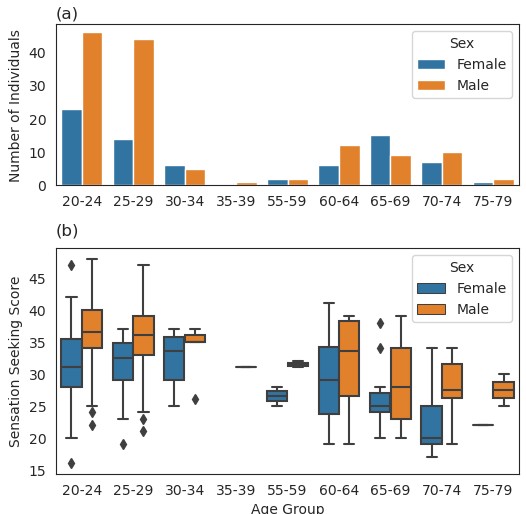

Fig. 1. Number of participants (a) and distribution of sensation-seeking scores (b) across different age and gender groups.

*3) MRI data acquisition:* Functional magnetic resonance imaging (fMRI) is a non-invasive neuroimaging technique used to study brain functions. fMRI measures brain activity by detecting the blood oxygen level-dependent (BOLD) signals that reflect changes in blood flow and oxygenated hemoglobin in response to neural activity. Resting-state fMRI (rs-fMRI) measures spontaneous brain activity during a state of rest without specific tasks or external stimuli. Rs-fMRI has been widely used to quantify temporal correlations between brain regions, known as brain connectivity, which relates to various physical and psychological processes in the human body [21]–[23]. In the LEMON dataset, rs-fMRI and structural MRI scans were obtained using a 3 Tesla scanner equipped with a 32-channel head coil. Participants were instructed to stay awake with their eyes open during rs-fMRI acquisition. The raw rs-fMRI data was preprocessed using Nipype [24], following

TABLE I
CANDIDATE MODELS FOR SELECTION AND OPTIMIZATION THROUGH
INNER CROSS-VALIDATION

| Model | Hyperparameters for tuning |
|---|---|
| Support Vector Regression | choice of the kernel, regularization parameter |
| Lasso | regularization parameter |
| Elastic Net | regularization parameter, ratio of $L1$ and $L2$ penalty |
| XGBoost | tree depth, booster, and loss reduction for node split |

the procedure provided by the LEMON dataset. For each individual, the preprocessing steps included: 1) discarding the first 5 slices to ensure stable signals; 2) 3D motion correction; 3) distortion correction; 4) rigid-body coregistration to each participant's anatomical image; 5) signal denoising; 6) band-pass filtering; 7) mean centering and variance normalization; 8) spatial normalization to MNI152 2mm standard space. After preprocessing, the time series extracted from each brain region were used to construct functional brain connectivity networks.

### B. Methods

*1) Brain functional connectivity:* Brain functional connectivity represents the temporal correlations of neural activity between pairs of brain regions. In this study, we utilized resting-state fMRI data to construct brain functional connectivity networks.

As illustrated in Figure 2 (a), to construct the brain functional connectivity network, we first partitioned the voxel-level fMRI data into regions of interest (ROIs) using the pre-defined Automated Anatomical Labelling (AAL) Atlas [25] to reduce the high dimensionality. In this study, 116 ROIs were identified with the AAL atlas. Next, we extracted the time series data from each ROI by averaging the BOLD signal across all voxels within the ROI. The resulting time series represents the neural activity within the ROI over time. We quantified the brain functional connectivity as pair-wise Pearson's correlation coefficient between the time series of all ROIs. This yields a $116 \times 116$ functional connectivity matrix for each individual, representing the brain connectivity network. Subsequently, we applied min-max normalization to each functional connectivity across participants to standardize the correlation values onto the same scale before fitting them into the model.

*2) Sensation-seeking score prediction framework:* We employed nested cross-validation for model selection and performance evaluation to predict sensation-seeking from functional connectivity, as illustrated in Figure 2 (b). Nested cross-validation combines an inner loop for model selection and hyperparameter tuning with an outer loop for unbiased model performance evaluation [26]. We adopted leave-one-subject-out cross-validation as the outer loop to evaluate the model's predictive performance and generalization ability across individuals. Within each training set of the outer cross-validation, we split the data into 10-fold cross-validation as the inner loop to optimize model selection and parameter settings.

To identify the optimal model for predicting sensation-seeking, we evaluated a variety of machine-learning models to capture either linear or non-linear relationships between functional connectivity and sensation-seeking. Support Vector Regression (SVR), Generalized Linear Model (GLM) - Elastic Net and Lasso, and Extreme Gradient Boosting (XGBoost) were considered as candidate models for their ability to effectively handle high-dimensional data and prevent over-fitting. These models provide a range of linear to non-linear approaches with increasing model complexities. The optimal model was selected through inner cross-validation. Table I shows the candidate models and the hyperparameters tuned via inner cross-validation.

*3) Effects of demographic factors in predicting sensation-seeking:* In order to understand how demographic features such as age and gender contribute to the prediction of sensation-seeking, we evaluated the sensation-seeking prediction framework under two scenarios with the entire dataset ($N = 205$):

(1) Integrated age and gender alongside functional connectivity data as predictors. We encoded gender as a categorical feature and age as a numerical feature (min-max normalized).

(2) Included only functional connectivity as predictors.

*4) Age-specific models:* **Baseline**: To demonstrate the need for age-specific modeling, we evaluated a baseline model from the literature that did not employ the age-specific approach. To our knowledge, Wan *et al.* 's study [16] is, to our knowledge, the only work predicting sensation-seeking from rs-fMRI. The authors used an Elastic Net model for prediction. To establish a baseline for comparison with our proposed age-specific modeling, we applied an Elastic Net-based prediction model to the entire male population of the LEMON dataset ($N = 131$).

In our proposed age-specific modeling approach, we partitioned the male population into different age groups and implemented the sensation-seeking prediction framework within each age group. By comparing the performance of these age-specific models against the baseline single-model approach, we aimed to demonstrate that age-specific models improve prediction accuracy and underscore the importance of considering age differences when modeling sensation-seeking.

*5) Evaluation metric:* Pearson's correlation coefficient $r$, $R^2$, Root Mean Squared Error (RMSE), and Normalized Root Mean Squared Error (NRMSE) were used to evaluate the predictive performance, which is calculated as follows:

$$r = \frac{\sum_{i=1}^{n}(y_i - \bar{y})(\hat{y}_i - \bar{\hat{y}})}{\sqrt{\sum_{i=1}^{n}(y_i - \bar{y})^2 \sum_{i=1}^{n}(\hat{y}_i - \bar{\hat{y}})^2}}$$

$$R^2 = 1 - \frac{\sum_{i=1}^{n}(y_i - \hat{y}_i)^2}{\sum_{i=1}^{n}(y_i - \bar{y})^2}$$

$$\text{RMSE} = \sqrt{\frac{\sum_{i=1}^{n}(y_i - \hat{y}_i)^2}{n}}$$

$$\text{NRMSE} = \frac{\text{RMSE}}{y_{max} - y_{min}}$$

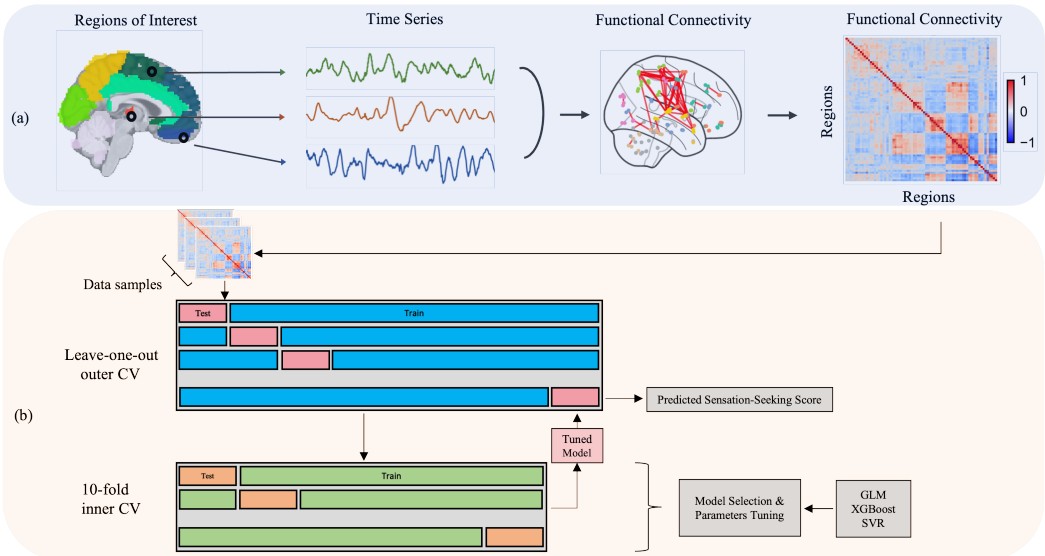

Fig. 2. Sensation-seeking score prediction from brain functional connectivity. The proposed framework comprised two steps: (a) The functional connectivity matrix was constructed as the pair-wise correlation of the BOLD signals extracted from brain regions; (b) The functional connectivities then served as input features of the prediction framework to predict sensation-seeking scores.

where $y_i$ is the actual sensation-seeking score, $\hat{y}_i$ is the predicted sensation-seeking score. $\bar{y}$ and $\bar{\hat{y}}$ are the mean values of the actual and predicted sensation-seeking scores, respectively.

*6) Feature importance:* we assessed the contribution of each functional connectivity feature across multiple model iterations within the leave-one-out cross-validation. To identify the most robust predictors of sensation-seeking, we focused on features consistently chosen by over 80% of the cross-validation folds. This approach ensured that the selected features had high predictive relevance across the individuals. For the selected predictors, we extracted the normalized coefficients from the fitted models to evaluate the importance of each predictor in predicting sensation-seeking.

## III. RESULTS

### A. Effects of demographic factors in predicting sensation-seeking

To assess the effect of demographic factors on sensation-seeking prediction, we evaluated the prediction framework under two scenarios: with and without age and gender as predictors. Both scenarios were assessed with the whole dataset of 205 participants.

Figure 3 shows the scatter plots of the predicted and actual sensation-seeking score with and without age and gender as predictors. Lasso regression was the optimal model in both scenarios. With age and gender included as predictors, the model achieved the accuracy of $R^2 = 0.30, \mathrm{RMSE} = 5.79, \mathrm{NRMSE} = 19.0\%$. After removing age and gender from the predictors, the prediction accuracy dropped to $R^2 = 0.13, \mathrm{RMSE} = 6.46, \mathrm{NRMSE} = 24.3\%$. We calculated the mean coefficient of age and gender across the fitted models as follows:

- Age: $\beta = -1.79$, $p = 5.50 \times 10^{-4}$, $\Delta R^2 = 0.13$
- Gender: $\beta = 3.08$, $p = 6.88 \times 10^{-5}$, $\Delta R^2 = 0.072$

where $\beta$ and $p$ denote the coefficient and p-value of each predictor in the linear regression model. $\Delta R^2$ shows the reduction in $R^2$ if a predictor is excluded from the model, which indicates its value added to the prediction. Our findings showed that including age and gender significantly increased the accuracy of sensation-seeking predictions. Age emerged as a significant predictor ($p = 5.50 \times 10^{-4}$); all else being equal, each additional year of age reduces sensation-seeking by 1.79 points. Gender also emerged as a significant predictor ($p = 6.88 \times 10^{-5}$); all else being equal, males have 3.08 points higher sensation-seeking scores than females. These findings align with existing literature that indicates age-related and gender-related variations in sensation-seeking [20].

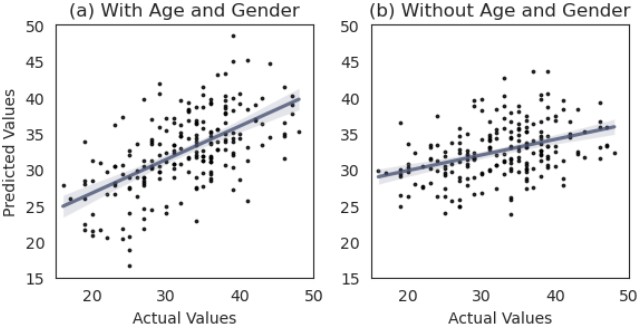

Fig. 3. Sensation-seeking prediction result with and without age and gender features ($N = 205$) through leave-one-out cross-validation. (a) The prediction with age and gender as predictors ($R^2 = 0.3$). (b) The prediction without age and gender as predictors ($R^2 = 0.13$).

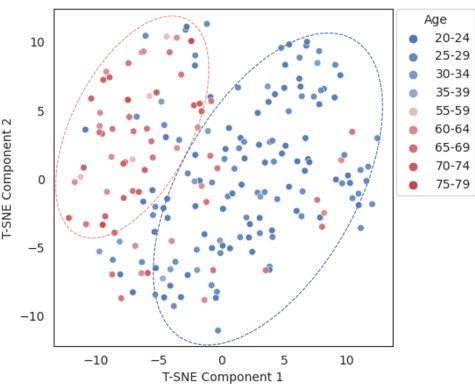

Fig. 4. t-SNE visualization of functional connectivity distribution across different ages, with each dot representing a data sample. Functional connectivity is distributed differently between young (depicted in blue) and older (depicted in red) age clusters.

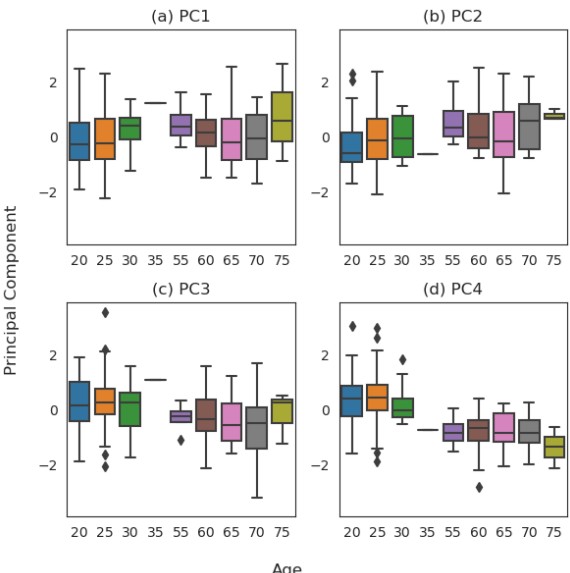

Fig. 5. Top four principal components distributions across different ages. Particularly, the distribution of principal component 2, 3, and 4 differs significantly across young (age $20 - 39$) and older (age $55 - 79$) age groups ($PC_2$: $p = 0.008$, $PC_3$: $p = 2.4 \times 10^{-4}$, $PC_4$: $p = 2.9 \times 10^{-13}$).

## B. Differences across age groups in predicting sensation-seeking

We computed the residuals between our model's predictions and the ground truth across different age and gender subgroups. The residual distribution varied significantly between the age groups $25 - 29$, $60 - 65$, and $70 - 75$ ($p < 0.05$). This variation suggested that the associations between functional connectivity and sensation-seeking may differ across different demographics. To further investigate the demographic-associated differences in brain functional connectivity, we applied the t-distributed Stochastic Neighbor Embedding (t-SNE) to visualize the functional connectivity distribution. As shown in Figure 4, the visualization shows distinct clusters corresponding to young (age $20 - 39$) and older (age $55 - 79$) age groups, indicating distinct functional connectivity distribution within young and older cohorts. Additionally, we performed Principal Component Analysis (PCA) on the functional connectivity data. Figure 5 shows the top four components (explained 45% of the variance), for instance. The principal components (PCs), as linear combinations of original functional connectivities, exhibited different distributions across young and older age groups, as shown in $PC_2$ ($p = 0.008$), $PC_3$ ($p = 2.4 \times 10^{-4}$), $PC_4$ ($p = 2.9 \times 10^{-13}$). These results show that functional connectivity is distributed differently across different aging phases. Therefore, a single predictive model may not be able to account for the age-related differences in functional connectivity patterns. Based on these observations, we hypothesized that developing separate models for each age subgroup would offer more accurate predictions by capturing the distinct neural patterns of sensation-seeking within each group.

## C. Predicting sensation-seeking within age groups

To test our hypothesis that fitting separate models for different age groups better captures the diverse neural correlates of sensation-seeking across various aging stages, we segmented the dataset into age subgroups. We focused on the male population due to its higher variability in sensation-seeking

scores and a larger sample size in each age group. Given the small number of participants in certain age ranges, we merged the age groups $30 - 34$ (5 participants) and $35 - 39$ (1 participant) with the $25 - 29$ group. Similarly, the age group $55 - 59$ (2 participants) was merged with the $60 - 64$ group, and the age groups $65 - 69$, $70 - 74$, and $75 - 79$ were combined to ensure sufficient data samples in each group. Then we developed predictive models tailored to each subgroup.

The baseline of the Elastic Net model fitted on the entire male group ($N = 131$) yielded prediction performance of $R^2 = 0.067, RMSE = 6.25$. Table II presents the best models and their respective performances for each subgroup, and Figure 6 illustrates the prediction results within each age subgroup. While Table III and Figure 7 compare the aggregated predictions from each subgroup to the baseline prediction result of the single-model approach. The results show a substantial improvement in prediction performance when fitting age-specific models to subgroups compared to the baseline model. The age-specific models achieved relatively high prediction performance in age groups $20 - 24$, $25 - 39$, and $65 - 79$ but failed to capture sufficient information in the $55 - 64$ age group. This result may be attributed to the smaller sample size and lower variability in sensation-seeking scores within this group, not allowing the model to capture the difference in functional connectivity as sensation-seeking changes.

## D. Functional connectivity associated with sensation-seeking within age groups

Our analysis revealed distinct associations between functional connectivity and sensation-seeking across different age

TABLE II
SELECTED MODEL AND EVALUATION METRIC OF THE MODEL FITTED IN EACH SUBGROUP (FC = FUNCTIONAL CONNECTIVITY)

| Participants (N: number of individuals) | Predictors | Model | $r$ | $R^2$ | RMSE | NRMSE |
|---|---|---|---|---|---|---|
| Male aged 20-24 (N=46) | FC | XGBoost | 0.56 | 0.31 | 5.03 | 19.0 |
| Male aged 25-39 (N=50) | FC | Lasso regression | 0.40 | 0.15 | 5.22 | 20.0 |
| Male aged 55-64 (N=14) | FC | SVR | -0.83 | -0.16 | 6.47 | 32.0 |
| Male aged 65-79 (N=21) | FC | Lasso regression | 0.67 | 0.44 | 4.84 | 19.0 |

TABLE III
SELECTED MODEL AND EVALUATION METRIC OF THE BASELINE AND AGE-SPECIFIC MODELS (FC = FUNCTIONAL CONNECTIVITY)

| Participants (N: number of individuals) | Predictors | Model | $r$ | $R^2$ | RMSE | NRMSE |
|---|---|---|---|---|---|---|
| Male population (N=131) | FC | Elastic Net regression (baseline) | 0.30 | 0.067 | 6.25 | 19.5 |
| Male population (N=131) | FC | Age-specific models | 0.62 | 0.38 | 5.12 | 18.0 |

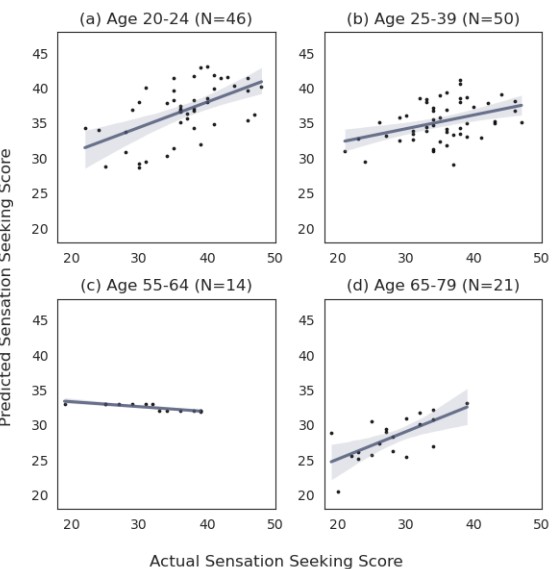

Fig. 6. Prediction result within each age group. (a) Prediction within age group $20-24$ ($R^2 = 0.31, p = 5.1 \times 10^{-5}$). (b) Prediction within age group $25-39$ ($R^2 = 0.15, p = 0.004$). (c) Prediction within age group $65-79$ ($R^2 = -0.16, p = 2.4 \times 10^{-4}$). (d) Prediction within age group $65-79$ ($R^2 = 0.44, p = 0.001$).

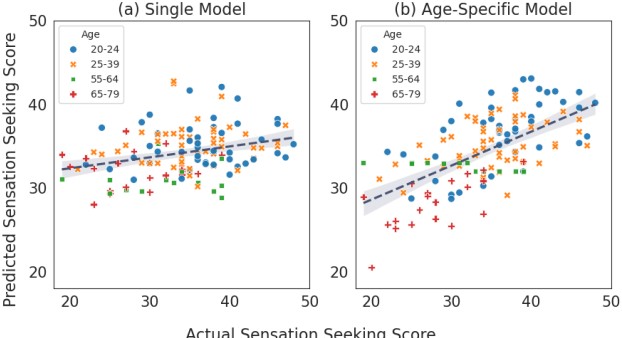

Fig. 7. Prediction result of the male population (N=131). Figure (a) shows the prediction from the baseline of the single-model approach ($R^2 = 0.067, p = 5.9 \times 10^{-4}$). Figure (b) shows the aggregated prediction from the age-specific approach that fits separate models within age subgroups ($R^2 = 0.38, p = 5.6 \times 10^{-15}$).

groups. To identify strong predictors of sensation-seeking, we determined the connectivities that were consistently selected by over 80% of the 131 cross-validation iterations. These robust predictors potentially provide valuable insights into the neural correlates of sensation-seeking. Table IV lists the selected functional connectivities within each subgroup and their normalized feature importance. Figure 8 shows the brain connectivity network in each group.

In the age group $20-24$, the XGBoost model, which captures non-linear relationships, provided the best predictive performance. The functional connectivities of hippocampus-superior parietal gyrus, cerebellum-thalamus, and cerebellum-paracentral lobule exhibited strong non-linear relationships with sensation-seeking. In the age group $25-39$, Lasso regression more accurately modeled the functional connectivity and

sensation-seeking associations. Nine functional connectivities were consistently selected by Lasso, among which amygdala-vermis connectivity was the strongest predictor of sensation-seeking. Within the age group $65-79$, the connectivities of middle frontal gyrus-amygdala, supramarginal gyrus-pallidum, and vermis-superior parietal gyrus, selected by Lasso regression, exhibited strong linear associations with sensation-seeking. Since the prediction accuracy in the age group $55-64$ was not promising, the functional connectivities selected in this group were not considered valid predictors.

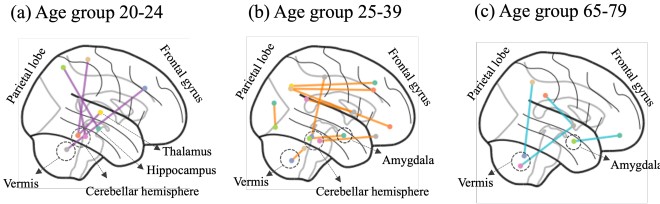

Fig. 8. Visualization of the selected brain connectivities within each age group denoted by different colors. The brain regions involve the parietal lobe, subcortical region, frontal gyrus, vermis, cerebellar hemisphere, thalamus, and cerebral hemisphere.

TABLE IV
SELECTED FUNCTIONAL CONNECTIVITIES WITHIN EACH AGE GROUP AND
THE NORMALIZED FEATURE IMPORTANCE

| Subgroup | Functional Connectivity Between Brain Regions | Feature Importance |
|---|---|---|
| Age 20-24 | Hippocampus R - Parietal Sup L | 0.26 |
| | cerebellum 4 5 R - Thalamus R | 0.18 |
| | cerebellum 4 5 R - Paracentral Lobule R | 0.12 |
| | Frontal Mid L - Vermis 9 | 0.08 |
| | cerebellum 3 R - Cingulum Post R | 0.04 |
| Age 25-39 | Amygdala L - Vermis 1 2 | 0.10 |
| | Occipital Mid R - Occipital Inf L | 0.07 |
| | Frontal Sup L - Angular R | 0.06 |
| | Frontal Med Orb R - Angular L | 0.06 |
| | Frontal Mid R - Angular L | 0.06 |
| | cerebellum 7b L - Vermis 10 | 0.06 |
| | cerebellum 3 L - Postcentral L | 0.06 |
| | Cingulum Post L - Caudate L | 0.06 |
| | Temporal Inf L - Rectus R | 0.05 |
| Age 55-64 | * | * |
| Age 65-79 | Frontal Mid Orb R - Amygdala R | 0.21 |
| | SupraMarginal R - Pallidum R | 0.12 |
| | Vermis 9 - Parietal Inf R | 0.1 |
| | cerebellum 7b L - Pallidum R | 0.09 |

## IV. DISCUSSION

### A. Comparison with existing study

In this study, we used functional connectivity derived from resting-state fMRI (rs-fMRI) to predict the sensation-seeking scores of 131 healthy male individuals in the LEMON dataset. Our predictive model achieved an $R^2$ value of 0.38 and RMSE of 5.12, indicating that sensation-seeking can be predicted from rs-fMRI with reasonable accuracy. The results obtained in our work suggest that brain connectivity patterns capture relevant information about the underlying neural mechanisms of sensation-seeking.

A previous study investigating sensation-seeking prediction from rs-fMRI reported a Pearson correlation $r = 0.34$ between the predicted and actual sensation-seeking scores in the cohort of 414 participants [16]. The study, however, did not acknowledge or address the variations in brain patterns associated with sensation-seeking across different demographic groups. Our analysis showed that functional connectivity is distributed differently across young and older age groups (Figure 4). The age-related variability in brain connectivity suggests that neural mechanisms underlying sensation-seeking may not be uniform between younger and older individuals, likely reflecting age-related neural changes. Therefore, a one-size-fits-all model is insufficient for sensation-seeking modeling in the all-ages population. By fitting age-specific models, we demonstrated that distinct models for different age subgroups offered more accurate predictions by capturing the diverse neural pattern associated with sensation-seeking across different aging phases (Figure 7).

### B. Brain functional connectivity linked to sensation-seeking

The selected functional connectivities that are associated with sensation-seeking included the brain areas such as prefrontal areas (e.g., middle frontal gyrus, superior frontal gyrus), cerebellum, subcortical regions (e.g., amygdala, hippocampus), parietal lobe (e.g., angular gyrus, supramarginal gyrus), and cerebral cortex areas (e.g., paracentral lobule), as listed in Table IV. The cerebellum is involved in cognitive functions, emotional regulation, and behavioral inhibition. Structural connections between the cerebellum and several cerebral cortex areas (including the precentral gyrus, paracentral lobule, precuneus, fusiform gyrus), thalamus, and putamen, are linked to individual novelty-seeking scores [27]. The prefrontal cortex area has a well-established role in impulse control and novelty detection. The interaction between prefrontal and subcortical regions such as the amygdala, striatum, and hippocampus have been found to be positively associated with novelty expectation and novelty seeking [28], [29]. The hippocampus is linked to novelty signal processing, which is involved in the dopaminergic neurotransmission that mediates the novelty and reward processing [30], [31].

Although previous studies have explored the brain regions identified in our work for their association with sensation-seeking, our study demonstrated that these associations are not uniform across all age groups. The brain connectivities associated with sensation-seeking vary significantly in different age groups. For instance, in the age group $20 - 24$, the association between brain connectivity and sensation-seeking score is non-linear, as captured by the XGBoost model. While in other groups, GLM captures the linear relationship between brain connectivity and sensation-seeking (Table II). These variations in functional connectivity indicate the complex neural interaction involved in sensation-seeking and highlight the importance of employing different models within each age group to identify these diverse neural patterns.

### C. Effects of normalization and age group division

To ensure the robustness of our age-specific results, we examined the effects of normalization and age group division on the prediction of sensation-seeking. First, to test whether the improved performance in age-specific models was due to per-group normalization, we normalized the functional connectivity data within each age group and evaluated the prediction framework on the entire male population. This approach yielded $R^2 = 0.001$, significantly lower than our proposed age-specific model. This result confirmed that the accuracy gain in our age-specific modeling was not from normalization. In addition, to determine whether the age group division was meaningful or random while yielding favorable results, we performed a permutation test. We randomly shuffled the ages of participants and ran the prediction framework 5 times. We did not observe consistent prediction patterns in these shuffled datasets, which demonstrated that the prediction results were not due to random chance or false positives.

### D. Limitation and future directions

A limitation of this study is the small sample sizes within the older age groups, especially in age groups $55 - 64$ and $65 - 79$. The limited data samples within these age ranges may limit the generalizability of our findings to older populations. On

the other hand, the younger age groups of $20-24$ and $25-39$ have larger sample sizes of 44 and 50 participants, respectively. In these subgroups, we demonstrated the impact of age on the neural patterns associated with sensation-seeking. Another limitation is we did not investigate gender differences in the neural correlates of sensation-seeking due to the limited number of female participants in each age group. Future research could include a larger and more gender-balanced dataset to explore the gender-related difference in neural mechanisms of sensation-seeking. We will also conduct a validation study on external datasets in future work to validate the generalizability of our findings, and provide a comprehensive investigation into the role age and gender play in the neural basis of sensation-seeking traits.

## V. CONCLUSIONS

Our study explored the predictive power of brain functional connectivity in predicting sensation-seeking and revealed the age-related difference in brain connectivity patterns associated with sensation-seeking. Our proposed age-specific prediction model achieved a prediction accuracy of $R^2 = 0.38$, considerably higher than the baseline that fitted a single model for the all-ages population ($R^2 = 0.067$). Our findings pointed out the limitations of the one-size-fits-all approach and demonstrated the necessity for subgroup-specific models that consider age differences in modeling sensation-seeking. Future research could extend this subgroup-specific modeling approach to other demographic groups, such as gender, to provide a comprehensive understanding of the neural correlates of sensation-seeking, and further explore more effective assessments and interventions for sensation-seeking behaviors.

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
