# OpenReview forum: "Predicting Sensation-Seeking from Resting-State fMRI: The Need for Age-Specific Models"
_IEEE.org/EMBS/BHI/2024/Conference — IEEE BHI'24_

### Official Review · Reviewer_qm9x · 2024-08-10
**Age-specific fMRI models predict sensation-seeking, revealing age-related brain differences**

**Overall Rating:** 7
**Confidence:** 3

**Other Quality Metrics:**

(a) Clarity of writing: great
(b) Clinical Significance: fair
(c) Methodological Novelty: great
(d) Experiments and Results: great

**Questions For The Authors:**

1. What challenges did you face when attempting to fill in the gaps in the chosen dataset, such as small sample sizes for older and female groups?
2.  Self-reported questionnaires are error-prone and biased, but they are simple, quick, and cheap whereas fMRI can be expensive and burdensome to perform. What clinical scenario would fMRI be used to evaluate sensation-seeking behavior?

**Strengths:**

1. The study introduces age-specific modeling for predicting sensation-seeking traits using resting-state fMRI data
2. The use of nested cross-validation for model selection and performance evaluation is a methodological strength. This approach enhances the reliability of the model predictions and helps prevent overfitting
3. The age-specific models significantly outperformed the baseline model that did not account for age differences, achieving a higher R2 value, suggesting the proposed method is more effective
4. The study successfully identifies key brain regions associated with sensation-seeking

**Summary Of The Paper:**

The paper explores the relationship between sensation-seeking behavior, which is a tendency towards seeking novel and stimulating experiences, and brain activity patterns using resting-state fMRI data. Sensation-seeking has implications for various behaviors like substance abuse or reckless driving, and traditionally, it has been measured through self-reports. The study aims to improve these measurements by using fMRI to examine brain functions associated with this trait. A key finding is that brain activity patterns related to sensation-seeking vary significantly with age. The researchers developed predictive models specific to different age groups, demonstrating that these tailored models are more accurate than a one-size-fits-all model used across all ages. They found that particular brain regions, including the prefrontal areas and the cerebellum, are notably involved in sensation-seeking behaviors, with these connections differing among age groups. By developing age-specific models, the researchers achieved better accuracy in predicting sensation-seeking scores from brain activity, indicating the importance of considering age when studying the neural bases of psychological traits.

**Weaknesses:**

1. The study focuses exclusively on a male population, which limits generalizability
2.  Small sample size within older age groups, especially 55-64 and 65-79
3. The paper acknowledges the influence of gender on sensation-seeking traits but does not explore these differences in depth due to the limited number of female participants. This is important because sensation-seeking and gender differences in brain connectivity and behavior are well-documented
4. While cross-validation is used, there is still a risk of overfitting since there isn't an external, independent test set used.
5. Without longitudinal data, the study cannot account for changes in sensation-seeking and brain connectivity over time
6. The axes labels for some figures, like Fig 5 and 7, are difficult to read. Figure 8 is not readable

---

### Official Review · Reviewer_BQjV · 2024-08-18
**A great novel research findings that could improve on some details in evaluations.**

**Overall Rating:** 7
**Confidence:** 3

**Other Quality Metrics:**

- Clarity of writing: **great**
- Clinical Significance: **great**
- Methodological Novelty: **good**
- Experiments and Results: **great**

**Questions For The Authors:**

N/A

**Strengths:**

- The manuscript presented clear connections between challenges, motivations, and proposed works.
- The manuscript employed several prediction/modeling techniques to comprehensively evaluate the impact of age-specific modeling.
- The manuscript presented good evaluations of its predictions and some interpretability.
- The manuscript shared important functional connectivities between brain regions and explained the causal connections.

**Summary Of The Paper:**

Sensation-seeking can reflect an individual's impulsivity and preferences for adventure-seeking and risk-taking. While recent studies investigated the plausibility of assessing sensation-seeking via functional magnetic resonance imaging (fMRI), they only evaluated generalized modeling and overlooked the influence of demographics. Therefore, the manuscript investigated the propensity of building age-specific sensation-seeking models and proposed a pipeline that contained data processing, and score prediction. It evaluated the proposed work on 131 male participants from the MPI Leipzig Mind-Brain-Body (LEMON) dataset and claimed overperformance over generalized modelings.

**Weaknesses:**

- **Evaluation**: It would be better to add Pearson's correlation coefficients on predicted vs. actual values to analyze the model's sensitivity.
- **Visualization**: The manuscript placed baselines, age-specific models, and aggregated ones altogether in Table II, making it difficult to follow and realize.

---

### Decision · Program_Chairs · 2024-09-23

Accept